# Prevalence of *Acinetobacter baumannii* Multidrug Resistance in University Hospital Environment

**DOI:** 10.3390/antibiotics14050490

**Published:** 2025-05-10

**Authors:** Francesco Foglia, Annalisa Ambrosino, Shahab Bashir, Emiliana Finamore, Carla Zannella, Giovanna Donnarumma, Anna De Filippis, Massimiliano Galdiero

**Affiliations:** 1Department of Experimental Medicine, University of Campania “L.Vanvitelli”, 80138 Naples, Italy; francesco.foglia@unicampania.it (F.F.); annalisa.ambrosino@unicampania.it (A.A.); shahab.bashir@unicampania.it (S.B.); carla.zannella@unicampania.it (C.Z.); giovanna.donnarumma@unicampania.it (G.D.); anna.defilippis@unicampania.it (A.D.F.); 2Complex Operative Unit of Virology and Microbiology, University Hospital of Campania “L.Vanvitelli”, 80138 Naples, Italy; emiliana.finamore@unicampania.it

**Keywords:** MDR *Acinetobacter*, antimicrobial resistance, carbapenem resistance, stewardship programs

## Abstract

**Background:** *Acinetobacter baumannii* is a significant pathogen and a major contributor to healthcare-associated infections, particularly in intensive care units. Its high potential for developing multiple drug resistance (MDR) makes it a challenging pathogen to manage. This study investigates the prevalence and resistance patterns of MDR *A. baumannii* isolates over a six-year period at a university hospital in Southern Italy. **Objective:** The aims of this study are to evaluate recent trends in the prevalence of MDR *A. baumannii*, analyze resistance patterns, and assess the impact of the antimicrobial diagnostic stewardship program implemented in 2018. **Methods:** This retrospective cohort study was conducted at the University Hospital of Campania “Luigi Vanvitelli” from 2018 to 2023. A total of 191 *A. baumannii* isolates from blood, urine, and wound samples were analyzed. Antimicrobial susceptibility testing was performed following EUCAST guidelines. The prevalence of MDR strains was assessed across three periods: pre-pandemic (2018–2019), during the pandemic (2020–2021), and post-pandemic (2022–2023) **Results:** Among the 191 isolates, 89.5% were classified as MDR. The highest number of isolates occurred in 2020, with blood cultures and urine samples increasing by 40.9% and 62.5%, respectively, while wound isolates decreased by 34.2%. The implementation of antimicrobial diagnostic stewardship programs correlated with a reduction in carbapenem resistance in 2020 and 2022. However, resistance to meropenem and colistin persisted. A 60.4% decline in total isolation from 2020 to 2023 suggests effective infection control measures. **Conclusions:** MDR *A. baumannii* remains a significant threat to healthcare. Although there have been slight reductions in resistance following antimicrobial stewardship interventions, persistent resistance to last-line antibiotics underscores the urgent need for alternative treatments, enhanced surveillance, and stricter infection control strategies.

## 1. Introduction

*Acinetobacter baumannii* is recognized as a challenging pathogen due to its high ability to infect hospitalized patients and its resistance to conventional antimicrobial treatments. This bacterium is often found in intensive care units (ICUs). It causes difficult-to-treat infections such as wound infections, bloodstream infections, and severe ventilator-associated bacterial pneumonia (VABP), particularly affecting immunocompromised individuals [1]. *A. baumannii* is classified as a high-alert human pathogen and ranks at the top of the critical priority list for developing new antibiotics. It is categorized as an *ESKAPEE* pathogen, alongside *Enterococcus faecium*, *Staphylococcus aureus*, *Klebsiella pneumoniae*, *Pseudomonas aeruginosa*, *Enterobacter* spp., and *Escherichia coli* [2,3]. Research shows that the death rate for ICU patients infected with *A. baumannii* ranges from 45% to 60%, and this rate can exceed 80% when these microorganisms show extensive drug resistance. The existing treatment options for addressing infections from multiple drug-resistant *A. baumannii* strains are insufficient. Public health faces a serious threat due to the rapid spread of multidrug-resistant (MDR) *A. baumannii*. This situation has worsened in recent years, with a continuous rise in MDR, extensively drug-resistant (XDR), and pandrug-resistant (PDR) hospital isolates of *A. baumannii* [4,5]. Some strains have even developed resistance to colistin, which is often used as a last-resort treatment [6].

The accurate identification of individual *Acinetobacter* species is crucial due to significant differences in antibiotic susceptibility and their clinical importance [7]. Matrix-Assisted Laser Desorption Ionization–Time of Flight Mass Spectrometry (MALDI-TOF MS) is the tool most used in diagnostic laboratories. It allows for the rapid and reliable identification of microorganisms. Its effectiveness relies on comparing protein spectra from clinical isolates with those stored in continuously updated commercial databases [8,9].

Notably, within the *Acinetobacter* genus, *A. baumannii* is recognized as the primary pathogen responsible for hospital-acquired infections worldwide.

Previously, *A. baumannii* was considered a minor pathogen. Still, it has proven to be an effective pathogen, causing opportunistic infections and possibly leading patients towards sepsis and multiple organ failure [10]. *A. baumannii* has recently emerged as a clinically significant pathogen responsible for ventilator-associated pneumonia and catheter-related infections, showing antibiotic resistance [11,12]. Furthermore, it is linked to various outbreaks of nosocomial infections, community-acquired infections, and infections related to war and natural disasters [2,11]. However, in intensive care units (ICUs), severe infections caused by this microorganism are the most prevalent. This has made *A. baumannii* a significant bacterial threat to human health and led to its classification as a critically important pathogen by the WHO [13]. Its ability to rapidly develop effective multidrug resistance (MDR) mechanisms and produce biofilms further worsen this threat [12,14]. This study aimed to accurately determine the prevalence rate of infections and antibiotic resistance level in *A. baumanii* clinical isolates in University Hospital of Campania. Bacterial strains were analyzed at the Complex Operative Unit of Virology and Microbiology over six years. The identified period was divided into three phases: pre-pandemic [2018–2019], during the pandemic [2020–2021], and post-SARS-CoV-2 pandemic [2022–2023].

## 2. Results

Throughout the monitoring period of 6 years, 10,742 blood cultures were performed, along with 10,503 urine tests, and 4693 samples that included pus, drainage, and swabs were collected. The count of *A. baumannii* was 82 [0.8%] in samples from the bloodstream, 30 [0.3%] in urine samples, and 79 [1,7%] in pus, drainage, and wound swabs for a total of 191 different strains isolated in 6 years. Considering a microorganism demonstrating resistance to three or more distinct classes of antibiotics as multidrug-resistant, we identified 171 [89.5%] MDR strains among a total of 191 isolated strains.

### 2.1. Characteristics of Acinetobacter Clinical Isolates

*A. baumannii* strains were isolated from 191 patients, 67% of them belonging to male patients, indicating a statistically significant male predominance [*p* < 0.05]. The sex ratio [M/F] was approximately 2.03. The highest number of *A. baumannii* isolation cases was recorded during the pandemic years [2020–2021], with a noticeable peak in 2020 (Figure 1).

A site sampling analysis of the *A. baumannii* isolates showed an increase in blood culture samples from 22 to 31 [+40.9%] and urine samples from 8 to 13 [+62.5%] during the pandemic. Conversely, isolates from pus, drainage, and swabs, which are typically familiar sources for *A. baumannii*, decreased significantly from 35 to 23 [−34.2%] compared to the pre-pandemic period [2018–2019] (Figure 2). This reduction was likely associated with decreased hospital admissions during the lockdown period. In the post-pandemic period [2022–2023], blood cultures and urine samples positive for *A. baumannii* showed fluctuations but remained lower than those in the 2020 peak, while the presence in pus, drainage, and swabs slightly increased in 2022 before dropping to only four strains in 2023.

The breakdown of isolates across different years demonstrated that the peak isolation occurred in 2020 with 48 strains, which gradually declined to 19 strains in 2023, reflecting a reduction of 60.4%.

### 2.2. Antibiotic Susceptibility Evaluation

In this study, *A. baumannii* was identified as one of the most prevalent multidrug-resistant microorganisms, showing a multidrug resistance (MDR) rate of 89.5%. After implementing the antimicrobial diagnostic stewardship program in 2018, a slight decrease in carbapenem resistance was noted in 2020 and 2022. However, many isolates in 2023 continued to show resistance to meropenem (Figure 3), indicating a potentially positive impact from ASP. However, the continued presence of carbapenem resistance underscores the ongoing challenge of antimicrobial resistance linked to this pathogen.

## 3. Discussion

*A. baumannii* has appeared as a significant nosocomial pathogen, posing a serious concern for the healthcare system due to its ability to develop resistance to multiple drugs [14]. Our study investigates the prevalence and resistance trends in *A. baumannii* at the University Hospital of Campania “Luigi Vanvitelli” over the past six years. It underscores the critical need for prompt surveillance programs, antimicrobial stewardship, and infection control measures to combat multidrug-resistant (MDR) strains. The peak isolation rates of *A. baumannii* were seen in 2020, coinciding with the COVID-19 pandemic, highlighting worldwide trends of secondary bacterial infections in immunocompromised individuals and patients with multiple health conditions. The spread of the COVID-19 pandemic has radically changed disease prevention protocols, including using masks, social distancing, stricter hand hygiene, and dedicated COVID-19 wards. All these preventive measures were highly effective in reducing the transmission of many community-acquired respiratory infection diseases. However, they were insufficient to prevent the dissemination of MDR hospital-acquired infections [15]. Indeed, clinical urgency and the lack of specific therapeutic options led to the widespread and inappropriate use of broad-spectrum antibiotics, such as azithromycin, chloroquine, and hydroxychloroquine [13]. The irrational use of these drugs generated a selective pressure that favored the spread of resistant strains, including *A. baumannii* [16,17].

In addition, the excessive use of invasive medical devices, particularly in ICUs, such as mechanical ventilators, central venous catheters, and urinary catheters, facilitated nosocomial transmission.

These factors contributed to the peak of MDR *A. baumannii* occurring during the COVID-19 pandemic, also showing the key role of strict antimicrobial stewardship measures during public health crises.

The prevalence of bloodstream infections and urinary infections caused by *A. baumannii* was recorded at 40.9% and 62.5%, respectively. Our findings align with those of Rawson TM et al., who noted that hospital-based infections were particularly prevalent during the pandemic due to prolonged hospital stays and increased exposure to multiple medical devices [13]. Another study was conducted at a Saudi hospital facility where the researchers found a considerable increase in *A. baumannii* infections in ICU patients [14,16]. In contrast to the previously mentioned isolations, wound and pus base isolations showed a decrease of 34.2%. This reduction may be attributed to fewer patients undergoing surgery and treatment for other diseases. The dual impact of the pandemic on nosocomial isolations underscores the need for robust infection control measures [18,19].

Our study found a consistently high prevalence of multidrug-resistant (MDR) *A. baumannii* at 89.5%. This finding highlights the bacteria’s ability to develop multiple-drug resistance [15,20]. These results are in line with previous reports, including a study from Pakistan, which noted increasing trends in *A. baumannii* prevalence and emphasized the necessity for regular surveillance to address the issue of drug resistance. Compared to 2020, a moderate decline in carbapenem-resistant bacteria was observed in 2022 and 2023, coinciding with the implementation of antimicrobial stewardship programs. Indeed, before this, the lack of ASP or diagnostic stewardship (DS) interventions contributed to the high and often inappropriate use of broad-spectrum antibiotics. The ASP and DS interventions adopted in 2018 include early detection supported by routine microbiological reports and the use of MALDI-TOF MS for rapid identification, as well as targeted therapy, promoted through regular audits by our multidisciplinary team, which provides feedback and recommendations on antimicrobial prescriptions. This integrated approach contributed to the observed decline in MDR isolates. Moreover, multiple studies support the idea that such stewardship can reduce the rate of antimicrobial resistance in hospital settings [21,22,23].

Despite radical measures, *A. baumannii* has shown resistance to the final line of drugs, including carbapenems and colistin. This carbapenem-based resistance is mainly due to the overexpression of the efflux pump [24].

Our study also highlighted that the prevalence of *A. baumannii* was higher in males than in females, with males representing 67% of the cases. These findings are consistent with another study, which suggests that various factors, such as biological differences, behavioral patterns, and access to healthcare facilities, may explain this discrepancy [1,25].

Colistin was once considered the last line of defense against multidrug-resistant (MDR) bacteria [26,27,28]. However, identifying certain *A. baumannii strains* resistant to colistin raises significant concerns, as it greatly limits the treatment options available for these MDR strains [29,30]. This resistance may result from lipid A modifications or the loss of lipopolysaccharides. Additionally, other methods of resistance development mentioned by Marino et al. include the upregulation of efflux pumps, enzymatic degradation, and modifications of outer membrane proteins [31].

The data presented in our study highlight the urgent need for alternative combinations of antimicrobial drugs to treat MDR *A. baumannii*. Options such as cefiderocol and sulbactam have demonstrated susceptibility against carbapenem-resistant *A. baumannii* [32].

The persistent high prevalence of *A. baumannii* can be attributed, in part, to its ability to form biofilms [33,34]. This process involves quorum sensing and activates additional resistance mechanisms, such as efflux pumps. These characteristics provide *A. baumannii* with protection against the host immune system, complicating treatment options. Research suggests that targeted strategies aimed at disrupting biofilms, such as quorum sensing inhibitors, could effectively manage *A. baumannii* infections [35].

This study has some limitations. First, there is no correlation between microbiological data and the clinical outcomes of the analyzed cases, such as morbidity, mortality, or treatment response. In addition, since this is a single-center study, our findings cannot be directly applied to other hospitals with different patient populations. Moreover, our data referred to the entire hospital, making it impossible to differentiate between COVID-19 and non-COVID-19 areas.

One positive outcome of our study was a 60.4% reduction in *A. baumannii* isolations from 2020 to 2023. This decline indicates that more effective control strategies for antimicrobial infections are being implemented. However, the decrease in resistance does not eliminate concerns within healthcare settings; hyper-resistant strains may still exist that are resistant to nearly all available antibiotics. Therefore, ongoing developments in resistance will continue to present significant challenges for healthcare facilities.

## 4. Materials and Methods

### 4.1. Study Period and Area

This retrospective study was carried out at the Unit Operative Complex of Virology and Microbiology of the University Hospital “L. Vanvitelli”, a hospital based in Naples. The hospital includes various medical departments, such as intensive care units and specialized units for treating infections. During the pandemic, the facility had a total capacity of 477 hospital beds. This study used electronic records from institutional laboratory information systems to collect data on patients admitted between 2018 and 2023 with positive samples for *A. baumannii*. This research examined demographic and clinical information, such as age, sex, hospital ward, isolated organisms, and corresponding antibiotic susceptibility profiles. Specimens were obtained from various sources, including blood cultures, pus, exudates, wound swabs, and urine. These sites were selected based on clinical relevance and the isolation frequency observed in our hospital setting. These sources are commonly associated with nosocomial infections in critically ill patients, especially those admitted to the ICU. Other sample types were excluded due to their low prevalence. For each patient, multiple samples that tested positive for *A. baumannii* within 96 h were consolidated into a single positive result.

### 4.2. Antimicrobial Stewardship Program

In 2018, the University Hospital “L. Vanvitelli” implemented an ASP to reduce the use of broad-spectrum antibiotics and the incidence of MDR bloodstream infections. This program was based on a prospective audit with feedback, during which a multidisciplinary team, comprising infectious disease specialists, microbiologists, pharmacists, and statisticians, carried out systematic evaluations of antibiotic prescriptions in the ICU three times a week and provided targeted recommendations on therapy choices. In parallel, a diagnostic stewardship program was supported by microbiologists, who routinely delivered local resistance reports to assist clinicians in evidence-based prescribing. This integrated, non-restrictive approach helped reduce the use of broad-spectrum antibiotics and the incidence of MDR bloodstream infections.

### 4.3. Identification and Sampling

The specimens were handled and cultured following global standards. Samples were prepared for culture using traditional, accepted techniques; cultures that tested positive underwent further scrutiny, and the species were identified using MALDI-TOF BioTyper 4.1 software [Bruker Daltonics, Bremen, Germany]. This was performed after isolating a single colony on MacConkey agar and incubating it overnight at 37 °C. The Phoenix Automated Microbiology System [Becton Dickinson and Company (BD), Franklin Lakes, New Jersey, USA] was used to conduct antimicrobial susceptibility testing [AST] integrated using the traditional Kirby–Bauer disk diffusion method, and the antimicrobial susceptibility of all the isolates was then evaluated following the Clinical and Laboratory Standards Institute (CLSI) antibiotic guidelines.

The Sensititre™ AST system [Thermo Fisher Scientific, Waltham, Massachusets, USA] was used to determine the Minimum Inhibitory Concentrations [MICs] for *ceftazidime–avibactam*, *ceftolozane–tazobactam*, *colistin*, and *piperacillin*–*tazobactam*. Additional tests were conducted for organisms resistant to multiple drugs, including the NG-Test CARBA 5 [NG Biotech, Guipry, France] and Xpert CARBA-R [Cepheid], specifically for organisms producing carbapenemase.

The interpretations of antimicrobial susceptibility testing [AST] and Minimum Inhibitory Concentrations [MICs] were conducted in compliance with the guidelines set by the European Committee on Antimicrobial Susceptibility Testing [EUCAST] [36].

### 4.4. Statistical Analysis

Data were examined using IBM SPSS 22.0 for IOS [IBM SPSS, Chicago, Illinois, USA]. Categorical variables were displayed as counts and proportions. The Chi-square test or Fisher’s exact probability method was used to compare qualitative data. A *p*-value < 0.05 signified a statistically significant difference.

## 5. Conclusions

Our findings emphasize the dynamic nature of *A. baumannii* infections, which are influenced by external factors such as the COVID-19 pandemic and internal interventions like antimicrobial stewardship programs. The consistently high prevalence of multidrug-resistant (MDR) strains, the emergence of colistin resistance, and the persistence of biofilms highlight the urgent need for new treatment strategies. Future research should focus on alternative antimicrobials, combination therapies, and the potential development of vaccines for *A. baumannii*.

A comprehensive approach is necessary to combat MDR *A. baumannii*, integrating ongoing surveillance, innovative antimicrobial solutions, and stringent infection control measures. Strengthening antimicrobial stewardship and investing in novel therapeutic strategies are critical in addressing the growing threat posed by this pathogen.

## Figures and Tables

**Figure 1 antibiotics-14-00490-f001:**
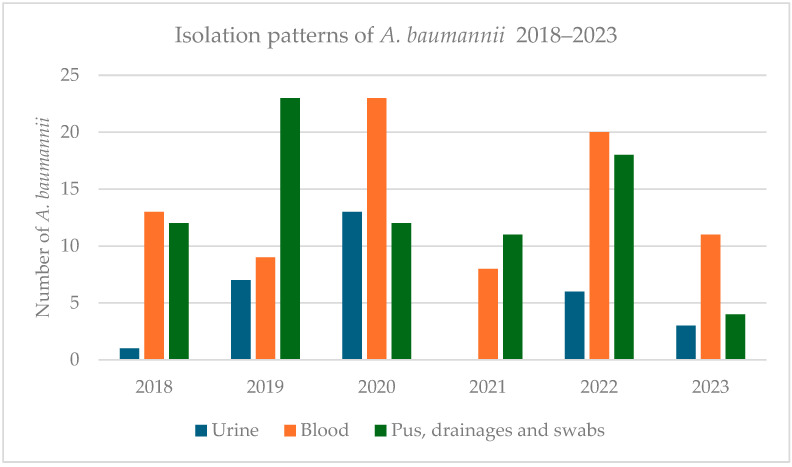
Isolation patterns of *A. baumannii* from various samples, including urine, blood, pus, drainages, and swabs.

**Figure 2 antibiotics-14-00490-f002:**
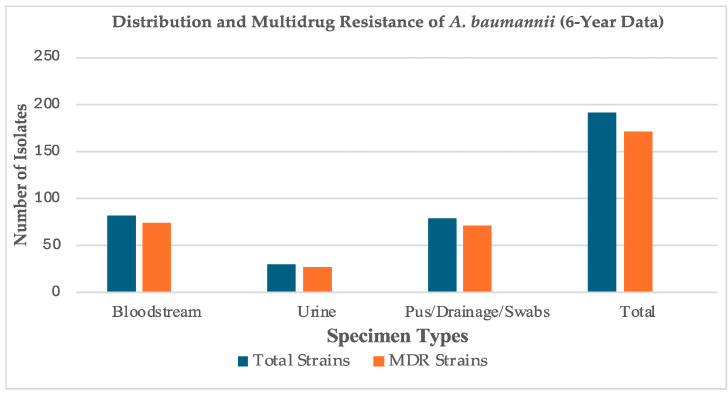
Distribution and multidrug resistance of *A. baumannii* strains across clinical specimens over 6 years.

**Figure 3 antibiotics-14-00490-f003:**
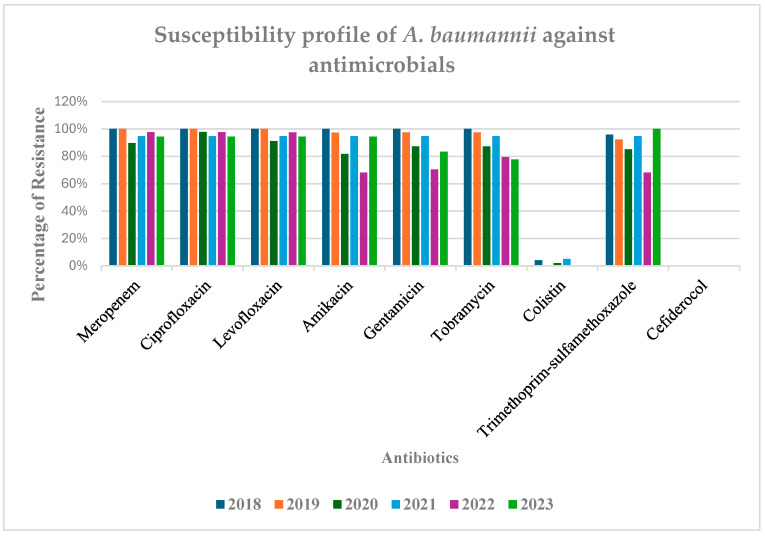
Percentage of bacterial resistance from 2018 to 2023 to following antibiotics: MER, meropenem; CIP, Ciprofloxacin; LVX, Levofloxacin; AMK, Amikacin; GEN, Gentamicin; TOB, Tobramycin; COL, colistin; TMP-SMX, Trimethoprim–sulfamethoxazole; FDC, cefiderocol.

## Data Availability

The original contributions presented in this study are included in the article/Appendix A. Further inquiries can be directed to the corresponding author.

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
