# Peer review of "Prevalence of Acinetobacter baumannii Multidrug Resistance in University Hospital Environment"

_antibiotics, 2025, doi:10.3390/antibiotics14050490_

Round 1
Reviewer 1 Report
Comments and Suggestions for Authors
Line 96: You mentioned ”a total of infected patients were observed”. What is this total? Are you referring to UTI, bloodstream infections, and wound infections? Are respiratory infections included in this total?
Line 121, 124: Can you explain what kind of Antimicrobial Diagnostic Stewardship Program was used in 2018? And also, the Antimicrobial Stewardship Program.
Line 187: What measures were taken to decrease the number of A. baumannii isolates?
Can you include whether it was early detection, targeted therapy, or enhanced infection control?
Do the authors consider the respiratory infection with A. baumanni relevant? It might be interesting to find the incidence of VAP with A. baumannii in the pandemic compared with the other two in the study.
A deeper discussion on the impact of the COVID pandemic is needed, as the paper describes a peak in 2020.
The authors should explain why they chose only those infection sites for their study. Could you clarify the inclusion criteria?
The authors should include a discussion on the Limitations of the study.
Author Response
1R. Line 96: You mentioned “a total of infected patients were observed”. What is this total? Are you referring to UTI, bloodstream infections, and wound infections? Are respiratory infections included in this total?
1A. Thank you for your observation. The missing number has been added. The total number refers only to UTIs, bloodstream infections, and wound infections. Respiratory infections are not included in the analysis because they did not fall outside the inclusion criteria. We clarified this point in the revised manuscript, paragraph 4.1 of the Materials and Methods section.
2R. Line 121, 124: Can you explain what kind of Antimicrobial Diagnostic Stewardship Program was used in 2018? And also, the Antimicrobial Stewardship Program.
2A. Thank you for your comment. In our University hospital, an Antimicrobial Stewardship Programme was implemented starting from 2018 to improve the appropriateness of antibiotic prescriptions (indication, choice of molecule, route of administration, duration of therapy). In this program a multidisciplinary team was identified, including infectious disease consultants, microbiologists, pharmacists, and statisticians, which developed intervention protocols and predefined outcome indicators. The intervention included periodic prospective audits (three times a week at regular intervals) conducted by two infectious disease consultants.
In parallel, a Diagnostic Stewardship team, including microbiologists, regularly provided reports on antimicrobial resistance patterns registered in our hospital, helping clinicians in making evidence-based decisions. We clarified this integrated approach in a new paragraph (named The Antimicrobial Stewardship Programme) of the Materials and Methods section.
3R. Line 187: What measures were taken to decrease the number of A. baumannii isolates?
3A. Thank you for your question. As clarified in the revised manuscript, the reduction in the number of A. baumannii isolates observed over the years appears to correlate with the implementation of AS and DS programs at our institution. Measures included: early detection using rapid diagnostic tools (MALDI-TOF MS), periodic audits of antimicrobial prescriptions by a multidisciplinary team, routine microbiological reporting of local resistance trends, and enhanced guidance for the empirical use of antibiotics.
4R. Can you include whether it was early detection, targeted therapy, or enhanced infection control?
4A. We thank the reviewer for the suggestion. The Antimicrobial Stewardship and Diagnostic Stewardship programs adopted at our University Hospital include early detection, supported by routine microbiological reports and the use of MALDI-TOF MS for rapid identification, and targeted therapy, promoted through regular audits by our multidisciplinary team, which provides feedback and recommendations on antimicrobial prescriptions. We included, as suggested, a brief description of these interventions in the Discussion section.
5R: Do the authors consider the respiratory infection with A. baumanni relevant? It might be interesting to find the incidence of VAP with A. baumannii in the pandemic compared with the other two in the study.
5A: We thank the reviewer for the valuable comment. We agree that the respiratory infections with A. baumannii are relevant, especially in critically ill patients during the COVID-19 pandemic, and would provide valuable insights to our study. For this reason, a retrospective study on respiratory tract infections related to A. baumannii is in progress.
6R: A deeper discussion on the impact of the COVID pandemic is needed, as the paper describes a peak in 2020.
6A: We thank the reviewer for the suggestion and have improved the discussion on the impact of the COVID pandemic (lines 154-168).
7R: The authors should explain why they chose only those infection sites for their study. Could you clarify the inclusion criteria?
7A: Thank you for your observation. Blood, urine, wound, drainage, and swab samples are the most common sources of nosocomial infection associated with A. baumannii. Other sites were excluded due to the insignificant number of isolations. We have added the inclusion criteria for selecting the infection site in paragraph 4.1 of the Materials and Methods section.
8R: The authors should include a discussion on the Limitations of the study.
8A: We agree with the reviewer’s assessment. The discussion has been implemented as requested (lines 222-227).
Reviewer 2 Report
Comments and Suggestions for Authors
General
This study analyzed 191 Acinetobacter baumannii isolates collected from a hospital in Naples between 2018 and 2023. Researchers found that 89.5% of the isolates were multidrug-resistant (MDR), with the highest number of cases during the COVID-19 pandemic in 2020. After implementing an antimicrobial stewardship program in 2018, there was a decline in incidence and prevalence of resistance (marginal). The study highlights the huge burden of MDR A. baumannii in Italy and the need for improved infection control, alternative treatments, and continued surveillance.
The study presents interesting and relevant data on the prevalence and resistance patterns of MDR A. baumannii, particularly in the context of antimicrobial stewardship efforts. The manuscript would nevertheless benefit from further refinements before publication.
Major suggestions
Introduction: no need to list all available typing methods when none was applied for the characterization of the isolates
Line 17 and 121: describe the elements of the AMS program introduced in 2018
Material and methods: The clinical setting is described only in terms of number of sample types, such as blood cultures and urine samples. Please also include additional hospital denominators, such as the number of beds, patient-days, and the percentage of COVID-19 patients during the pandemic years.
Minor suggestions
Throughout the manuscript: once Acinetobacter baumannii is introduced, use A. baumannii on all other occasions.
Throughout the manuscript: once “multidrug-resistant” is introduced, use “MDR” on all other occasions.
Line 70: use a comma after Previously
Line 78: correct to “This has made Acinetobacter baumannii a significant bacterial threat to human health and led to its classification as a critically important pathogen…”.
Line 96: a total of ……. – the number is missing
Line 106: probably it should be 2022 ?
Line 123: one punctuation mark too much.
Figure 3: Acinteobacter spp. or A. baumannii ?
Line 144: the sentence is not correct, please include the A. baumannii.
Line 180: no need to include the CRAB acronym at this point of manuscript.
Line 191: change immune to resistant
Comments on the Quality of English LanguageThe manuscript needs to be improved, and the language checked due to several typographical mistakes and inconsistencies.
Author Response
General
This study analysed 191 Acinetobacter baumannii isolates collected from a hospital in Naples between 2018 and 2023. Researchers found that 89.5% of the isolates were multidrug-resistant (MDR), with the highest number of cases during the COVID-19 pandemic in 2020. After implementing an antimicrobial stewardship program in 2018, there was a decline in incidence and prevalence of resistance (marginal). The study highlights the huge burden of MDR A. baumannii in Italy and the need for improved infection control, alternative treatments, and continued surveillance.
The study presents Interesting and”rele’ant data on the prevalence and resistance patterns of MDR A. baumannii, particularly in the context of antimicrobial stewardship efforts. The manuscript would nevertheless benefit from further refinements before publication.
Major suggestions
1R. Introduction: no need to list all available typing methods when none was applied for the characterization of the isolates
1A: We thank the reviewer for the observation. The paragraph has been revised to include only the identification and characterization methods that were used.
2R.Line 17 and 121: describe the elements of the AMS program introduced in 2018
2A. Thank you for your valuable suggestion. We clarified this implemented approach in a new paragraph, named The Antimicrobial Stewardship Programme in the “Materials and Methods” and “Discussion” sections.
3R. Material and methods: The clinical setting is described only in terms of number of sample types, such as blood cultures and urine samples. Please also include additional hospital denominators, such as the number of beds, patient-days, and the percentage of COVID-19 patients during the pandemic years.
3A. We thank the reviewer for this valuable observation. Unfortunately, detailed hospital denominators, such as the percentage of COVID-19 patients during the pandemic years, were not available at the time of data collection. However, we have included in the Material and methods section the number of beds available at University Hospital of Campania "Luigi Vanvitelli" during the pandemic period.
Minor suggestions
4R: Throughout the manuscript: once Acinetobacter baumannii is introduced, use A. baumannii on all other occasions.
4A: We modified the manuscript as suggested by the reviewer.
5R: Throughout the manuscript: once “multidrug-resistant” is introduced, use “MDR” on all other occasions.
5A: We thank the reviewer for this suggestion and have modified the manuscript.
6R. Line 70: use a comma after Previously
6A: We have done.
7R. Line 78: correct to “This has made Acinetobacter baumannii a significant bacterial threat to human health and led to its classification as a critically important pathogen…”.
7A: We thank you for the correction. We have done.
8R. Line 96: a total of ……. – the number is missing
8A. Thank you for your observation. The missing number has been added.
9R. Line 106: probably it should be 2022?
9A. Thank you for your comment. The correct year is 2022, and the sentence has been corrected.
10R. Line 123: one punctuation mark too much.
10A: We have corrected the punctuation mark.
11R. Figure 3: Acinteobacter spp. or A. baumannii ?
11A. The graph is related to the percentage of A. baumannii resistance to different antibiotics. We have corrected the title.
12R. Line 144: the sentence is not correct, please include the A. baumannii.
12A: We modified the sentence as suggested.
13R. Line 180: no need to include the CRAB acronym at this point of manuscript.
13A: The acronym CRAB has been removed.
14R. Line 191: change immune to resistant
14A. Thanks for the suggestion, we modified the text as requested.
Comments on the Quality of the English Language
R. The manuscript needs to be improved, and the language checked due to several typographical mistakes and inconsistencies.
A. We appreciate the reviewer’s recommendation regarding language improvement. We have carefully revised the manuscript to correct typographical errors and inconsistencies, ensuring that the language now aligns with the standards of scientific writing.
Round 2
Reviewer 1 Report
Comments and Suggestions for Authors
The revisions done to the manuscript are satisfactory.
Author Response
Thank you for your positive comments